# Terahertz and mid-infrared plasmons in three-dimensional nanoporous graphene

Fausto D'Apuzzo[1,2], Alba R. Piacenti[3], Flavio Giorgianni[3], Marta Autore[3], Mariangela Cestelli Guidi[4], Augusto Marcelli[4], Ulrich Schade[5], Yoshikazu Ito[6], Mingwei Chen[6,7,8] & Stefano Lupi[9]

Two-dimensional (2D) graphene emerged as an outstanding material for plasmonic and photonic applications due to its charge-density tunability, high electron mobility, optical transparency and mechanical flexibility. Recently, novel fabrication processes have realised a three-dimensional (3D) nanoporous configuration of high-quality monolayer graphene which provides a third dimension to this material. In this work, we investigate the optical behaviour of nanoporous graphene by means of terahertz and infrared spectroscopy. We reveal the presence of intrinsic 2D Dirac plasmons in 3D nanoporous graphene disclosing strong plasmonic absorptions tunable from terahertz to mid-infrared via controllable doping level and porosity. In the far-field the spectral width of these absorptions is large enough to cover most of the mid-Infrared fingerprint region with a single plasmon excitation. The enhanced surface area of nanoporous structures combined with their broad band plasmon absorption could pave the way for novel and competitive nanoporous-graphene based plasmonic-sensors.

[1] Istituto Italiano di Tecnologia and Dipartimento di Fisica, Università di Roma La Sapienza, Piazzale Aldo Moro 2, I-00185 Roma, Italy. [2] Advanced Light Source Division, Lawrence Berkeley National Laboratory, Berkeley, California 94720, USA. [3] INFN and University of Rome La Sapienza, Department of Physics, P.le A. Moro 2, 00185 Rome, Italy. [4] INFN-LNF, via E. Fermi 40, 00044 Frascati, Italy. [5] Helmholtz-Zentrum Berlin fur Materialien und Energie GmbH, Methoden der Materialentwicklung, Albert-Einstein-Strasse 15, 12489 Berlin, Germany. [6] WPI Advanced Institute for Materials Research, Tohoku University, Sendai 980-8577, Japan. [7] School of Materials Science and Engineering, Shanghai Jiao Tong University, Shanghai 200030, China. [8] CREST, Japan Science and Technology Agency, Saitama 332-0012, Japan. [9] CNR-IOM and Department of Physics, University of Rome La Sapienza, P.le A. Moro 2, 00185, Rome, Italy. Correspondence and requests for materials should be addressed to S.L. (email: stefano.lupi@roma1.infn.it).

Plasmons, the collective oscillations of electrons in metals and doped semiconductors, show notable properties including a strong interaction with the electromagnetic field, reduced wavelength in comparison with that of exciting light and a huge electric field enhancement several orders of magnitude larger than the incident light field. Those properties are at the basis of surface-enhanced Raman and Infrared spectroscopies which are now routinely used in many bio-sensing applications[1–3]. However, in looking for tunable and active devices, plasmonics has seen a major breakthrough since the discovery of graphene[4,5]. Indeed, due to the unique properties of this material, combining massless Dirac fermions, charge-density tunability, high electronic mobility, optical transparency and mechanical flexibility, graphene plasmons are a promising ingredient for smart devices, with applications across many fields, such as opto-electronics, photo-detectors and bio-sensing[6]. Furthermore, the low dimensionality of graphene induces an extreme compression of plasmons[7], especially at mid-infrared (MIR) and terahertz (THz) frequencies, which allows their nanoscale confinement, as it has been theoretically predicted and further experimentally demonstrated[8,9]. With these motivations, many graphene micro- and nano-structures have been proposed and realized in the last few years, combining different shapes and sizes[5,10,11], multi-layer systems, different substrates and hybrid devices, including active configurations for plasmonic control through optical[12] and electrical pulses[4,8,9].

Recently a novel fabrication process allowed researchers to obtain a 3D nanoporous configuration of high-quality monolayer graphene[13–17], composed by a 3D network of interconnected graphene monolayer with bicontinuous porosity. This nanoporous structure provides an extremely enhanced effective surface area and a third dimension to graphene. While nanoporous graphene (NPG) is object of ongoing studies to explore its applications for energy collecting electrodes[14,15], high-efficiency steam generation by solar illumination[16] and Li-air batteries[17], very little is known about its electronic and optical properties. This work, to the best of our knowledge, is the first investigation of the optical behaviour of nanoporous graphene by means of THz and MIR spectroscopy. Here we show the presence of intrinsic 2D Dirac plasmons in 3D NPG disclosing their behaviour with controllable doping level and tunable porosity. By taking into account the enhanced surface area of these nanoporous structures[18] in combination with their tunable plasmons, this work paves the way for innovative graphene-based plasmonic-sensors.

## Results

**Optical Measurements.** Several samples of 3D NPG were grown by nanoporous Ni-based chemical vapour deposition (CVD) method[13,19] (see Methods section). The as-grown samples (showing an intrinsical n-type conductivity and an average Fermi energy $E_F \sim 70$ meV) have a thickness spanning from 10 to 30 μm and their pore size can be engineered from about 200 nm to nearly 1 μm. The samples here investigated were well characterized through photoemission (PES), Raman spectroscopy, and transport measurements[13]. These measurements revealed that the 3D architecture of NPG preserves the 2D graphene character such as massless Dirac fermions and high electron mobility. A measure of the pore average size was obtained with the Barrett-Joyner-Halenda (BJH) method[13–16]. The Fermi energy can be further increased by doping nanoporous graphene with Nitrogen atoms (substitution of around 4% Carbon atoms), obtaining N-doped NPG (N-NPG), with values of $E_F$ up to 340 meV (ref. 13).

The optical transmittance $T(v)$ was measured from the THz range (30 cm$^{-1}$) to the Near Infrared (7,000 cm$^{-1}$) at room-temperature with a Michelson Bruker 66v Interferometer kept under vacuum. Due to their micrometric thickness the samples of NPG are free-standing so they have been directly mounted onto a metal frame located in the focal plane of a Michelson interferometer. A typical extinction spectrum $E(v) = 1 - T(v)$ of a NPG sample having an average porous size $p = 200 \pm 50$ nm and a thickness $t = 10 \pm 1$ μm, is reported in Fig. 1c. Single-layer graphene exhibits in the near-IR and visible range a constant universal absorption value of $\pi\alpha \sim 2.3\%$, and therefore a transmittance $T = 97.7\%$. This high-frequency absorption is related to Dirac intercone transitions. NPG samples instead, due to their finite (micrometric) thickness show, in the same spectral range, a nearly constant transmittance as low as a few percent or lower corresponding to an extinction $E \sim 99\%$ (Fig. 1c). From the measured extinction we finally obtain the real part of the optical conductivity $\sigma(v)$ by means of a Kramers–Kronig consistent fit (see Methods section). This quantity, shown in Fig. 1d, is normalized to its high frequency value $\sigma_{HF}$ evaluated around 7,000 cm$^{-1}$. Three main features are well evident in Fig. 1d: above 3,000 cm$^{-1}$, $\sigma(v)$ increases saturating at higher frequency. This absorption, like in single-layer graphene, can be associated with intercone electronic transitions. The small narrow peak around 1,600 cm$^{-1}$ roughly coincides with the graphene G phonon which becomes infrared active probably due to disorder. In the following we do not discuss this spectroscopic feature. At low frequency, where for a single layer non patterned graphene having a finite Fermi energy one would expect a Drude absorption, NPG instead exhibits a peak at a finite frequency. This peak, which has been also observed in single-layer disordered graphene[20], will be attributed (see below) to localized surface plasmons in the underlying graphene layers, where the localization and extra wavevector are provided by the nano-scale porosity. This observation is evidence of graphene plasmons in a 3D porous nanostructure.

**Analytical model.** The plasmon and the interband transition are separated by a broad minimum as the Dirac intercone electronic transitions are forbidden for photon energies below the Fermi energy (Pauli-blocking). Thus to separate the plasmon absorption from the interband transition, and tracing the plasmon characteristic frequency versus the Fermi energy and the pore size, we fit the optical conductivity through the sum of two components: a Lorentz oscillator, which describes the low-frequency peak, plus the conventional graphene interband term[21,22]:

$$\sigma(v) = \frac{Bv^2 \Gamma_{pl}}{\left(v_{pl}^2 - v^2\right)^2 + \left(\Gamma_{pl} v\right)^2}$$
$$+ C\left[\tanh\left(\frac{hv - 2E_F}{4k_B T}\right) + \tanh\left(\frac{hv + 2E_F}{4k_B T}\right)\right] \quad (1)$$

here $v_{pl}$ and $\Gamma_{pl}$ are the plasmon frequency and linewidth, respectively, while $T$ and $E_F$ are the temperature and the Fermi energy. $B(C)$ defines the intensity of the plasmon absorption (interband transition). To the terms of equation (1) one should add a flat background (with values $A \sim 0.3$–0.4), depending on the sample, already observed in single layer and few layers graphene, whose origin is probably related to charged impurities and extrinsic scattering centers[20,23]. As the phonon spectral weight is very small, we do not add the phonon (Lorentzian) contribution to equation (1).

The fit of the optical conductivity experimental data to equation (1) data are shown in Fig. 1d through a blue solid curve. For any reasonable fitting parameter choice, the interband transition and in particular its edge, cannot be well described by the previous model. Indeed, as described by equation (1), the

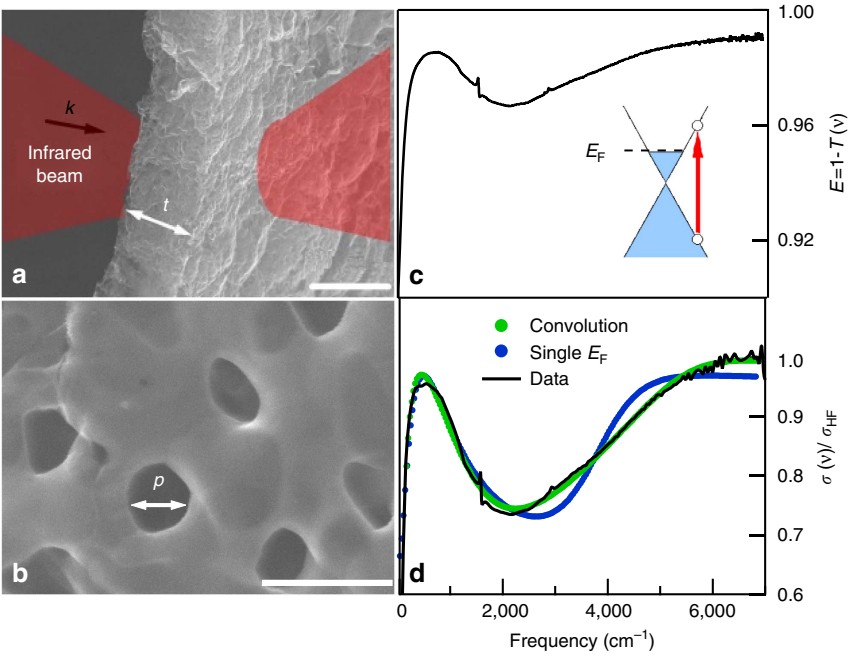

**Figure 1 | Nanoporous graphene structure and optical results.** SEM images at lower (**a**) and higher (**b**) magnifications of a NPG sample with a thickness $t = 10 \pm 1\,\mu m$ and average pore size $p = 200 \pm 50\,nm$. Scale bars correspond to 10 μm and 500 nm for (**a,b**) respectively. In (**a**) we also show the optical transmittance measurement scheme. (**c**) Typical extinction spectrum exhibits, at high frequency, a saturated absorption and, at low-frequency, a plasmonic broad peak separated by the Pauli-Blocking minimum. In the inset of (**c**) we show schematically a Dirac interband transition (red arrow). (**d**) Optical conductivity as extracted from extinction data (**c**), by a Kramers-Kronig consistent fit, normalized to its high-frequency value (black solid curve) compared to models with a single (blue solid curve) and distributed (green solid curve) Fermi energy (Supplementary Note 2). SEM, scanning electron microscope.

interband transition has a step-like behaviour at $2E_F$, whose slope increases for decreasing temperature. For the NPG sample shown in Fig. 1 the thermal smearing of the interband threshold at 300 K is nearly $1,000\,cm^{-1}$ against an experimental broadening of about $3,000\,cm^{-1}$. This suggests that both disorder and a spatial inhomogeneity of Fermi Energy should be taken into account in order to describe the experimental data as suggested also for single layer graphene[23]. Furthermore, spatial variations in the optical conductivity couldn't be resolved with far-field methods down to 35 μm spot size (Supplementary Note 1) and might arise at the micron- or sub-micron scale. Therefore, in order to fit more properly the experimental optical conductivity, we convolve the second term in equation (1) with a Fermi energy distribution function. Assuming for simplicity a flat distribution between two extremal values, $E_{F1}$ and $E_{F2}$, the convolution yields an analytical expression for $\sigma(v)$ (Supplementary Note 2). The resulting fit, which describes much better the optical data, is shown through a green solid curve in Fig. 1d. Let us observe that the modelling of the interband contribution is robust against the choice of the Fermi enery distribution function. Indeed, other distributions like the gaussian, provide a similar result although the convolution with the second term in equation (1) cannot be analytically calculated (see Supplementary Fig. 2 for the comparison between the flat and gaussian convolution fit).

The convolution model thus provides information on both absorption mechanisms (plasmon and interband), allowing one to retrieve the relevant parameters defining the plasmon behaviour: the plasmon frequency $v_{pl}$ and the average Fermi energy $E_F = (E_{F1} + E_{F2})/2$ (Supplementary Table 1). This last quantity as extracted from the fitting is in good agreement with that directly measured on samples of the same batch.

## Discussion

In micro- and nano- ordered structures based on conventional metals and 2D electron gas systems[3,5,24–29], the plasmon frequency depends on both the charge-carrier density and the geometrical parameters of the structure. In a disordered system one still expects a dependence on these parameters, although disorder may affect both the plasmon frequency, linewidth and intensity[30].

In this section we study the optical conductivity of nanoporous graphene samples as a function of charge-density, that is, of the Fermi energy, for a fixed pore size, which allows one to investigate the effect of doping for a given geometry. We span a Fermi energy interval from 70 meV, for as-grown NPG samples, to 340 meV in intentionally Nitrogen doped systems (N-NPG)[31,32]. As the whole 3D graphene structure does not appreciably change as a consequence of Nitrogen doping, the optical properties of pure and N-NPG samples can be usefully compared.

In Fig. 2 (from a to d) we show the SEM images of samples having the same average porosity ($p = 200 \pm 50\,nm$) and thickness ($t = 10 \pm 1\,\mu m$), versus an increasing Fermi energy. The corresponding optical conductivity normalized to its high-frequency value is plotted in the right panels (from e to h). The Fermi energy values were obtained from fitting to the experimental optical conductivity (see above) and correspond well to the values provided by independent measurements on samples from the same batch[13]. The whole fit, the plasmon, and the interband components are represented by blue-dashed, blue-dotted and green-dotted lines, respectively (see Supplementary Table 1 for values of fitting parameters). When $E_F$ increases towards higher energies (from e to h), the plasmon peak is blue shifted. This is expected when the carrier density increases due to a stiffening of the plasmonic restoring force[4].

The plasmon frequency $v_{pl}$ versus $E_F$ as obtained from fitting of data in Fig. 2, is reported in Fig. 3 (red squares). The error bars have been estimated through a statistical analysis on several samples having, nominally, the same properties (see Supplementary Note 3). $v_{pl}$ clearly increases with $E_F$ passing from nearly $60\,cm^{-1}$ for $E_F \sim 70\,meV$ to about $800\,cm^{-1}$ for the

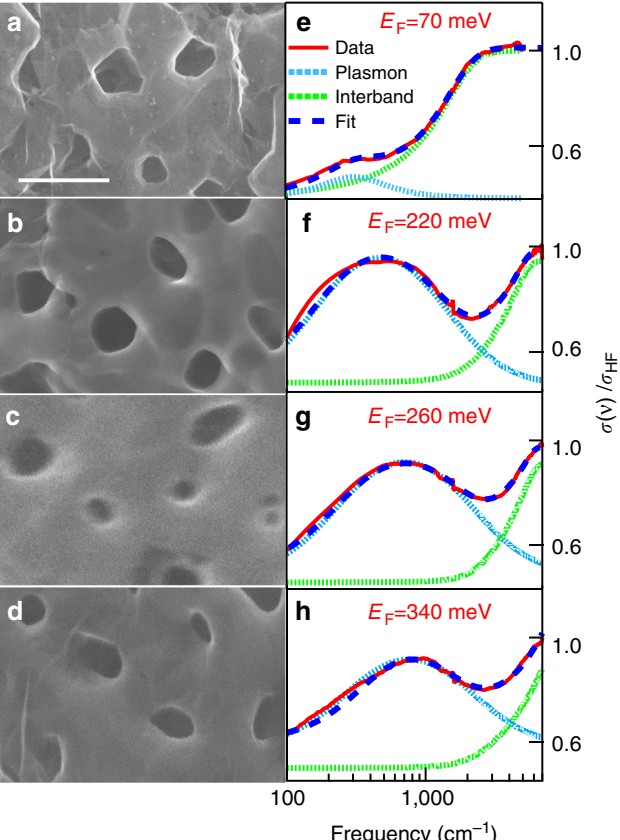

**Figure 2 | Optical properties of nanoporous graphene at fixed pore size for different Fermi energies.** (**a**–**d**) SEM images of nanoporous graphene samples with the same average pore size $p = 200 \pm 50$ nm, and an increasing Fermi energy (from **a**–**d**). The scale bar corresponds to 500 nm. (**e**–**h**) Optical conductivity curves with relative value of Fermi energy. The whole fitting, the plasmon and the interband components (see text) are represented by blue-dashed, blue-dotted and green-dotted lines, respectively. SEM, scanning electron microscope.

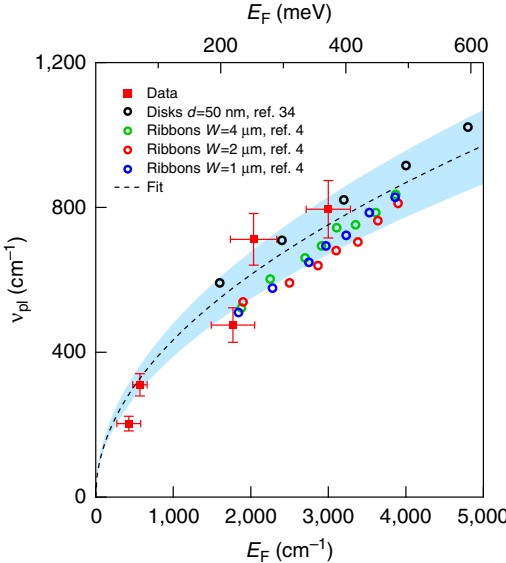

**Figure 3 | The plasmon frequency versus the Fermi energy.** Data (red square markers) are compared with a fit based on Dirac two-dimensional (2D) plasmon frequency equation (equation (2), black dashed line) (see text and Methods section). The shaded blue area represents the effect of parameter variation on the fit. The good match among experimental data and fit permits one to assign the plasmon mode in NPG and N-NPG samples to 2D Dirac carriers. The black (coloured) open circles represent data from refs 4,34 for ribbons and disks, respectively. Disk diameter $d$ and ribbons width $W$ are reported in data labels. The agreement among literature and experimental data reinforces the assignment of plasmons in 3D NPG in terms of Dirac 2D charge carriers. NPG, nanoporous graphene; 3D, three dimensional.

highest doped N-NPG sample with $E_F \sim 340$ meV. The value of $\Gamma_{pl}$ also increases from 200 to 2,500 cm$^{-1}$ with increasing $E_F$, and is much larger than the mean values found in 2D graphene[4]. Let us notice, that the mobility of Dirac carriers in the same NPG samples have been recently measured in ref. 33 providing quite good values ranging from 5,000 to 7,500 cm$^2$ V$^{-1}$ S$^{-1}$. Therefore, although we cannot completely rule-out a possible increase of losses in NPG plasmons with respect to single-layer graphene, we mainly ascribe their broad linewidth $\Gamma_{pl}$ to the statistical distribution of Fermi energy and pore size in the 3D structure (Supplementary Note 4).

As discussed in the introduction (see also ref. 13), 3D nanoporous graphene can be considered as of an incoherent superposition of single-layer graphene. This suggests that one may compare its plasmon behaviour with that of a single layer system. More specifically, we make a comparison among the experimental plasmon frequency in 3D nanoporous graphene with the theoretical one as obtained by an analytical calculation (see Methods section) for a 2D square array of circular holes in a single layer of graphene[34]:

$$h\nu_{pl} = \sqrt{\frac{2\alpha\hbar cLE_F}{\pi(\epsilon_1 + \epsilon_2)\bar{p}}} \qquad (2)$$

Here $\alpha \approx 1/137$ is the fine-structure constant, $c$ is the speed of light, $\hbar = h/2\pi$ is the Plank constant, $\epsilon_1 = \epsilon_2 = 1$ are the permittivity of

the two interfacing media (vacuum in our case) and $L$ is a geometrical factor[34]. The value of $L$ depends on the geometrical shape of the micro/nano object (disk, ring, ribbon), and has been set to 12.5 for a perfect circular disk composed of a single layer graphene. However, one could expect a variation of $L$ with the disk shape (circular versus elliptical) and on the graphene thickness, which has been fixed to 0.5 nm in ref. 34. Due to pore shape distribution (from circular to elliptical), and thickness variation of graphene single-layer in 3D graphene (especially around the pore[15]), we use $L$ as a free fitting parameter, while all other parameters in equation (2) have been determined by the present experiment. By fitting equation (2) to data, one obtains $L = 10.5 \pm 0.5$ and the corresponding fit is reported through a black-dashed line in Fig. 3. The error bars (represented by a blue shaded area around the dashed black line), have been estimated through a statistical analysis which takes into account the uncertainties on both $E_F$, $p$ and $L$. As evident from Fig. 3 the theoretical 2D plasmon frequency is in very good agreement to experimental data, indicating that the plasmonic behaviour of 3D graphene can be well described in terms of Dirac 2D plasmons.

To further support this finding, we have compared our data to experimental observations available in recent literature, which accounts for plasmons in graphene structures ranging in a variety of sizes (from 50 nm to 4 µm), shapes (disks and ribbons) and interfaces. In particular we superimpose on our data in Fig. 3, the results on single-layer graphene nano-disk (black open circles) as measured in ref. 34, and those on single-layer graphene micro-ribbons (coloured open circles) as measured in ref. 4. In order to have an effective comparison, literature data have been renormalized, using equation (2), to account for different size and dielectric environment (see Methods section for details).

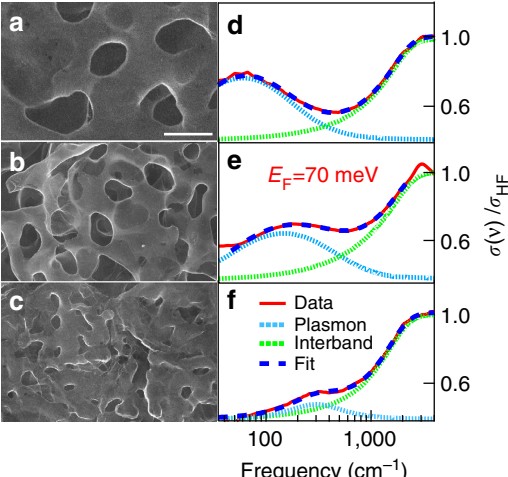

**Figure 4 | Optical properties of NPG at fixed Fermi energy for different pore size.** (a,b) SEM images of nanoporous graphene samples with the same average Fermi energy $E_F \sim 70$ meV, and decreasing pore size. Scale bar corresponds to 1 µm. (d–f) corresponding optical conductivity curves (pore size decreases from a–c). The whole fitting (see equation (1) in the text), and the plasmon and interband components are represented by blue-dashed, blue dotted and green dotted lines, respectively. NPG, nanoporous graphene; SEM, scanning electron microscope.

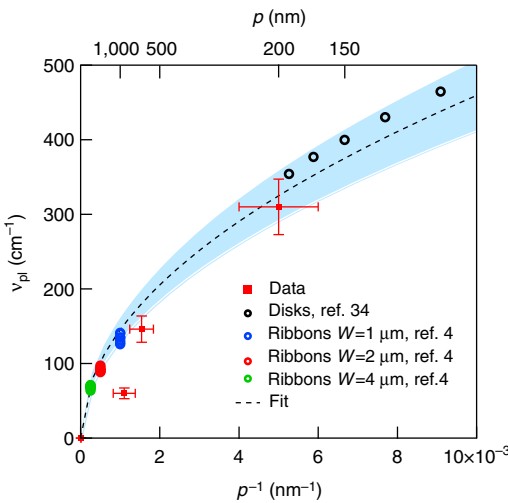

**Figure 5 | The plasmon frequency versus the inverse of pore size $p^{-1}$.** NPG experimental points (red markers) are compared with analytic calculation (black dashed line). The shaded blue area represents the effect of parameter variation on the calculation. Black empty circles (colored circles), correspond to plasmon frequency as measured in an array of nano-disks (nano-ribbons) which are rescaled to NPG data according to equation (2) to account for different Fermi energy and dielectric environment (see Methods section for details). NPG, nanoporous graphene.

The good comparison among experimental data, theoretical fit and literature reports suggests that, although we are dealing with a disordered 3D structure, the plasmon excitation in NPG and N-NPG preserves a 2D Dirac character. This indicates further that plasmonic modes in NPG and N-NPG graphene represent the natural extension in 3D of Dirac plasmons in single-layer/few-layers graphene.

We also investigated the optical conductivity of nanoporous graphene samples as a function of the pore size for a constant Fermi energy. This allows one to extract the dependence of the plasmon frequency on the pore size at a fixed charge density.

In Fig. 4, we show both the SEM images (from a to c) and the optical conductivity (always normalized to its high-frequency value, from d to f), in three different NPG samples having, nominally, the same average Fermi energy $E_F \sim 70$ meV and an average pore size of $900 \pm 100$, $650 \pm 90$ and $200 \pm 50$ nm, respectively. As evident in SEM images, pore sizes are randomly distributed. While the statistical pore spatial distribution mirrors in spectra independent of light polarization (Supplementary Note 5), from Fig. 4 it is possible to recognize a systematic optical-absorption pore size dependence. Indeed, while the interband transition falls in the same frequency range for all samples, the plasmon peak blue-shifts for decreasing pore size as expected by plasmonic thumb-rules[30]. From the fitting procedure (equation (1) and Methods section), we were able to extract the plasmon frequency $\nu_{pl}$ whose behaviour versus $1/p$ ($p$ is the average pore size), is plotted in Fig. 5. The increase of $\nu_{pl}$ from 80 to 300 cm$^{-1}$ shows an appreciable dependence on the pore size. Experimental error bars have been estimated through a statistical analysis on several samples having, nominally, the same properties. The behaviour of $\nu_{pl}$ versus $1/p$ can be calculated by using the same analytic model summarized in the previous section (equation (2)). Here we calculate $\nu_{pl}$ as a function of $1/p$ for $E_F = 70$ meV and $L = 10.5 \pm 0.5$. The calculation (black dashed line) well describes the experimental data. The error bars on the theoretical data (represented by a blue shaded area around the dashed black line), take into account the uncertainties on the parameters entering the calculation. The good agreement between

data and theory further allows us to attribute this spectral feature to plasmon excitation of Dirac carriers in NPG.

In Fig. 5, we also plot for comparison the behaviour of the plasmon frequency in an array of single-layer graphene nano-disk (black open circles)[34] and micro-ribbons (coloured open circles)[4], as in Fig. 3. To isolate the dependence on $p$, frequencies from literature data have been rescaled according to equation (2) to account for different Fermi energy and dielectric environment (see Methods for details). As in Fig. 3, the good agreement among our data and literature data in 2D graphene further reinforce the assignment of plasmon in NPG to massless Dirac carriers.

In conclusion, this work represents a systematic optical investigation of 3D nanoporous structures made of high-quality single-layer graphene. The optical conductivity of this non-periodic array of graphene nano- and micro-pores exhibits both single-carrier interband transitions and collective plasmonic modes resonating at Terahertz and Mid-Infrared frequencies. The plasmonic excitation depends both on doping and on the nanostructure geometry, the latter providing the extra-momentum needed to activate the radiation absorption process. The plasmon frequency dependence on the charge-carrier density (parametrized in terms of the Fermi energy), and pore size are in good agreement with a 2D Dirac character of plasmonic excitations in the 3D architecture of nano-porous graphene. This suggests that the extreme wavelength-compression and field enhancement of graphene plasmons is at work. Furthermore for N-doped samples, the microscopic inhomogeneity of doping and pore sizes yields a macroscopic plasmonic response that covers a wide spectral range ($> 1,000$ cm$^{-1}$). By taking into account the enhanced surface area of nanoporous structures, the tunability of graphene plasmon and the broad spectral response, the use of 3D NPG could pave the way for novel and competitive graphene based plasmonic-sensors.

## Methods
**Preparation of nanoporous graphene by CVD.** Ni$_{30}$Mn$_{70}$ ingots were prepared by melting pure Ni and Mn (purity $> 99.9$ at.%) using an Ar-protected arc melting furnace. After annealing at 900 °C for 24 h for microstructure and composition

homogenization, the ingots were cold-rolled to thin sheets with a thickness of $\approx 50\,\mu m$ at room temperature. Nanoporous Ni substrates were prepared from these sheets, by chemical dealloying them in a 1.0 M $(NH_4)_2SO_4$ aqueous solution at 50 °C. After de-alloying, the Ni substrates were rinsed thoroughly with water and ethanol and dried in vacuum.

Nanoporous Ni substrates loaded in a quartz tube were inserted into the center of a quartz-tube furnace and annealed at 800 or 900 °C under 2,500 sccm Ar and 100 sccm $H_2$, respectively. After the reduction pre-treatment, benzene (0.5 mbar, 99.8%, anhydrous) or pyridine (0.2 mbar, 99.8%, anhydrous) was introduced with the gas flow of Ar (2,500 sccm) and $H_2$ (100 sccm) for graphene growth. Typical CVD time is 2 min cleaning and 2 min deposition time at 800 °C for 200 nm pore size, 6 min cleaning and 2 min deposition time at 900 °C for 600–700 nm pore size and 18 min cleaning and 2 min deposition time at 900 °C for 1 μm pores. Moreover, typical CVD time of N-doped graphene is 2 min cleaning and 2 min deposition at 800 °C for 200 nm pore size, 8 min cleaning at 900 °C and 2 min deposition time at 800 °C for 600–700 nm pore size and 20 min cleaning at 900 °C and 2 min deposition at 800 °C for 1 μm pores. The furnace was quickly opened to quench the inner quartz tube with a fan to room temperature. The nanoporous Ni substrates were dissolved by 1.0 M HCl solution and then transferred into 2.0 M HCl solution to dissolve the residual Ni and Mn. The samples were repeatedly washed in distilled water for 5 times and floated on water for overnight.

**Microstructure characterization and physical property measurements.** The microstructure of the NPG samples was characterized by a scanning electron microscope (SEM, JEOL JSM-6700). The chemical bonding states of the Nitrogen doped NPG were studied by X-ray photoelectron spectroscopy (XPS, AXIS ultra DLD, Shimazu) with Al Ka and X-ray monochromator. Raman spectra were recorded by using a micro-Raman spectrometer (Renishaw InVia RM 1,000) with an incident wavelength of 514.5 nm. Finally, Photoemission spectroscopy was used for investigating the electronic properties of nanoporous graphene around the Fermi energy[13,14].

**Optical properties of nanoporous graphene.** We measured the optical transmittance of 3D NPG samples from the THz range (30 cm$^{-1}$) to the Near Infrared (7,000 cm$^{-1}$). Measurements were performed with a Michelson Interferometer Bruker 66v kept under vacuum and at room-temperature. The sample was fixed onto a copper frame with a 2 mm aperture, allowing the film to be free-standing over the aperture. An empty copper frame with an identical aperture was used to collect the reference spectrum $I_0(v)$. Transmittance $T(v)$ was then computed as the ratio $I_S(v)/I_0(v)$ between the spectrum transmitted by the sample $I_S(v)$ and the reference.The real part of the optical conductivity $\sigma(v)$ was obtained from the measured $T(v)$ by means of a Kramers-Kronig consistent analysis software (RefFit by Kuzmenko, A., available at https://sites.google.com/site/home-pageofalexeybkuzmenko/software).

**Plasmonic dispersion model.** Analytic calculation of the electromagnetic behaviour of NPG can be obtained considering NPG and N-NPG like an incoherent superposition of graphene single layers decorated by circular holes. The polarizability of a hole of diameter $p$ can be written for p$\omega \ll c$, ($\omega = 2\pi v$), as:

$$\alpha = p^3 \frac{A}{(2L)/(\epsilon_1 + \epsilon_2) - i\omega p/\sigma(\omega)} \quad (3)$$

Here, $A$ and $L$ are material- and size-independent constants[34], whereas $\sigma(\omega)$ is the graphene conductivity which takes the following form:

$$\sigma(\omega) = \frac{e^2}{\pi\hbar^2} \frac{iE_F}{\omega + i\gamma} \quad (4)$$

where $e$ is the elementary charge, $\hbar$ is the reduced Plank's constant $\gamma$ is the scattering rate of Dirac electrons.

By substituting equation (9) in equation (8), the plasmon frequency can be determined by looking for the zero of the real part of the polarizability. This corresponds to:

$$\hbar\omega_{pl} = h v_{pl} \sim e \left[ \frac{2\alpha\hbar\, cLE_F}{\pi(\epsilon_1 + \epsilon_2)p} \right]^{1/2} \quad (5)$$

as reported in the text, where $\alpha$ is the fine-structure constant. Furthermore interlayer plasmonic hybridization can be confidently neglected given that the expected energy decay of graphene plasmons is extremely compressed and estimated in the 5 nm range[5], much smaller than the porosity length scale typical of NPG, so that the underlying graphene layers can be assumed plasmonically decoupled. According to this relation plasmonic resonances in different graphene structures reported in literature can be quantitatively compared to NPG given that appropriate scaling factors are used to account for different $E_F$, $p$, $L$ and dielectric environments $\epsilon_1$ and $\epsilon_2$.

**Data availability.** The data that support the findings of this study are available from the corresponding author on request.

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

## Acknowledgements

This work was partly sponsored by JST-CREST 'Phase Interface Science for Highly Efficient Energy Utilization'; and the fusion research funds of 'World Premier International (WPI) Research Center Initiative for Atoms, Molecules and Materials', MEXT, Japan. We acknowledge BESSY II for providing infrared synchrotron radiation.

## Author contributions

Y.I. and M.C. fabricated and characterized nanoporous graphene samples. F.D., A.R.P., F.G., M.A., M.C.G., A.M., U.S. and S.L., carried out the terahertz and Mid-Infrared experiments and data analysis. F.G. and A.R.P. performed the theoretical calculations. F.D. and S.L. planned and managed the project with inputs from all the co-authors. F.D. and S.L. wrote the manuscript. All authors extensively discussed the results.

## Additional information

**Competing interests:** The authors declare no competing financial interests.

**Publisher's note**: 

