## [Peer Review File · Nature Communications]

Reviewers' comments:

Reviewer #1 (Remarks to the Author):

The authors report on plasmon resonance in 3-dimensional nanoporous graphene. The resonance frequency ranges from 200cm⁻¹ to 1000cm⁻¹, depending on the Fermi energy and pore size. From the dispersion, the 2d nature of the plasmons is verified. The 3d nature of the structures would allow novel type of sensors.

In principle, this work is interesting and relevant for future applications of graphene plasmonics. However, some relevant discussion and details are missing:

*The advantage of 3d graphene could be the enhanced surface area compared to 2d graphene. It would be helpful to discuss quantitatively the strength of the plasmon resonance in terms of absorption enhancement relative to the interband absorption. For 2d graphene, the plasmon-mediated absorption is enhanced by a few factors relative to the interband absorption (e.g. enhanced from 2.3% to ~30%). How strong is this enhancement for 3d graphene?

This point is related to the polarisability, and a comparison between 3d and 2d would be useful. Also the values for the polarisability, extracted from the fits are missing.

*The fit parameters should be given in the text and/or captions. What is found for gamma? Also here a comparison is welcome: e.g. for 3d graphene gamma is larger (more losses) but then there are more charges (stronger oscillator strength). How do these two factors balance?

*Minor: caption Fig 1c mentions black arrow but panel shows red arrow

Reviewer #2 (Remarks to the Author):

In this letter, the authors report on the observation of infrared plasmon in nanoporous graphene samples. This reviewer thinks that these claims are unsubstantiated in the present manuscript for the reasons below, therefore the manuscript cannot be published in Nature Communications.

1) The authors compare their structure to monolayer graphene. However the geometry they have is more closely resembles amorphous carbon films; i.e. random graphite. The thickness of the films are ~10 um in which case it is highly unlikely that all the "graphene" monolayers making up the thick film are electronically decoupled and behave optically and electronically as independent monolayers.

2) The structures are completely random leaving many geometric parameters hard to determine. Although the authors perform electron microscopy to determine the pore sizes, the areas characterized in the IR spectrometer can be very different. These two measurements are very difficult to co-localized given the complicated setups required for both techniques.

3) The most troubling aspect is that all the conclusions are based on fitting the experimental results based on some mathematical model that depend on many parameters such as the pore size, the Fermi level, the film thickness etc. All of these parameters may vary significantly from one area of the sample to another due to the random nature of this specific material system. It is not immediately obvious to this reviewer as to why the fits are unique given that there maybe another set of parameters to produce the same results. In fact it is also not clear why the particular mathematical form is used for the fits. In fact in line 193 they argue that a Gaussian can be used instead of the Lorentzian they adapted.

4) The authors claim that the resonances originate from the pores in a conducting "graphene" layer implying that they emerge as localized surface plasmon resonances. It is not clear in the text what they mean by the term plasmon. Is it volume plasmons, surface plasmons, or localized

surface plasmons? If it is the first, then it is not surprising since any conducting material including metals and doped semiconductors behave as a good reflector below the plasma frequency. If it is surface plasmons—a mode supported at the graphite and air interface, usually they cannot be excited from free space due to the momentum mismatch. If it is localized surface plasmons, the pores are 3D in nature and vary significantly from area to area and layer to layer on the samples based on the SEM images.

5) Even if we were to assume that the monolayers of graphene in the sample were electrically decoupled, optically they are only less than nm apart and strongly coupled. This is expected to affect the spectral position of the localized surface plasmon resonances significantly. In this case it is not clear why a model based on monolayer is adapted.

Reviewer #3 (Remarks to the Author):

The manuscript studies the optical properties of 3D nano porous graphene. They found that it exhibits plasmonic resonances in the mid-infrared, and perform systematic studies of the scaling behavior of this plasmonic resonance with pore size and doping. I find the study very well executed, and the results well explained and presentation very clear. To best of my knowledge, I think this is the first such study, and its results should be very useful for the development of 3D nano porous graphene for various applications. I recommend this work for publication in your journal if the below can be adequately addressed.

- the abstract should be further strengthened, particularly highlighting the impact of this study for applications in the midinfrared. Unlike graphene plasmonics, here the 3D nature implies that the plasmonic effect is polarization insensitive and can be used more broadly in certain type of applications.

- would the plasmon to plasmon hybridization be important in the analysis of the plasmon resonance. I would expect this to be sensitive to the filling factor. In other words, what is the pores volume to the total volume.

- the observed plasmon line width appears to be quite large. The authors should discuss the source of "damping" or probably this is due to pore size variability.

- the data set in figure 3 and 5 is quite scarce, where most of the data points were taken from other published work. I feel more experiment data should be included in figure 3 and 5. With more data, they can also analyze the plasmon line width and analyze its dependence with pore size. Line width should increase with decreasing pore size, per what has been reported in nanoribbons array.

Reply Reviewer 1

Point 1) The advantage of 3d graphene could be the enhanced surface area compared to 2d graphene. It would be helpful to discuss quantitatively the strength of the plasmon resonance in terms of absorption enhancement relative to the interband absorption. For 2d graphene, the plasmon-mediated absorption is enhanced by a few factors relative to the interband absorption (e.g. enhanced from 2.3% to ~30%). How strong is this enhancement for 3d graphene? This point is related to the polarisability, and a comparison between 3d and 2d would be useful. Also the values for the polarisability, extracted from the fits are missing.

Reply 1

While plasmons in 2D graphene exhibit an extinction peak from a few percent up to 25%, plasmonic excitation measured in 3D nanoporous graphene show an enhanced extinction, with values greater than 95% (see Fig.1 of the main manuscript). This enhancement is ascribable both to the large effective number of layers that constitute the 3D graphene configuration and on NPG microscopic polarizability.

In order to quantitatively discuss the plasmon strength and its ratio with the interband absorption, we will compare the real part of the optical conductivity both for single-layer and 3D based plasmons. The strength of the plasmon peak in the optical conductivity (i.e. its spectral weight), is also a measure of the corresponding polarizabilities.

In Fig. 1 of the present reply we show both the NPG plasmon band (for the NPG sample characterized by $p=200$ nm and $E_F=340$ meV, see Fig 4a of the main manuscript), and for a single layer plasmon peak representative of data available in literature (corresponding to an extinction peak of 12%, to a linewidth of $\Gamma=120$ cm^{-1} , and to a central frequency of 750 cm^{-1}). The real part of the optical conductivity of these bands is normalized to their interband high frequency part σ_{HF} .

The plasmon band in the optical conductivity provides a similar spectral weight (area under the plasmon peak) in both cases. This demonstrates that oscillator strengths of plasmons in NPG are comparable to that found in single-layer graphene although distributed over a larger spectral range

due to geometrical and doping inhomogeneity (see Reply 2). The broader footprint of NPG plasmons could be a useful feature when considering applications in Surface Enhanced Infrared Absorption.

Fig. 1: Infrared conductivity spectra of nanoporous graphene (red) and monolayer graphene (black), normalized to the interband high-frequency value σ_{HF} . Central frequency for both plasmons is around 750 cm^{-1} . The black dashed vertical lines represents the minimum and maximum 3D plasmon frequency as due to statistical width in both E_F and p (see Reply 2).

Point 2) *The fit parameters should be given in the text and/or captions. What is found for gamma?*

Also here a comparison is welcome: e.g. for 3d graphene gamma is larger (more losses) but then there are more charges (stronger oscillator strength). How do these two factors balance?

Reply 2

In Table S1 of SI (see below), we report the fit parameters related to data in Fig.2 and 4 of the main manuscript. In the first column, NP and NPN are relative to undoped and N-doped samples respectively. The II, III, IV and Vth columns report the pore size p , the plasmon frequency and widths ν_{pl} , Γ_{pl} , with the average Fermi energy E_F and the corresponding statistical width. Let us note that data reported in Table S1 correspond to an average on several samples having, nominally, the same physical properties.

Sample	p (nm)	ν_{pl} (cm ⁻¹)	Γ_{pl} (cm ⁻¹)	E_F (cm ⁻¹)
NP1	200 ± 50	280 ± 30	440 ± 45	570 ± 150
NP2	900 ± 100	60 ± 5	190 ± 20	580 ± 250
NP3	650 ± 90	150 ± 15	480 ± 50	560 ± 300
NPN1	200 ± 50	450 ± 50	1550 ± 150	1770 ± 600
NPN2	200 ± 50	670 ± 70	2550 ± 250	2100 ± 600
NPN3	200 ± 50	800 ± 80	2450 ± 250	2800 ± 1000

Table S1. Physical parameters for the measured nanoporous samples.

As evident from the Table, NPG plasmons show a large spectral broadening Γ_{pl} . This is actually larger than linewidth related to single layer graphene plasmon located at similar frequency. For instance, the low-doped NP2 sample shows a THz plasmon centered around 150 cm⁻¹ with a spectral width Γ_{pl} of about 480 cm⁻¹. THz plasmons in single-layer graphene decorated with disk or ribbon array, with a similar central frequency, presents a linewidth around 100 cm⁻¹ [see Ref. 4 of the main manuscript and Yan et al., Nature Nanotech. 7, 330 (2012)]. A similar overdamped behavior can be observed in Table S1 also for mid-IR plasmons.

In our opinion, this behavior is not related to an intrinsic low-mobility of Dirac carriers in 3D NPG graphene. Indeed, the carrier mobility has been recently measured in similar NPG samples [Y. Tanabe et al., “Electric Properties of Dirac Fermions Captured into 3D Nanoporous Graphene Networks” Advanced Materials, 2016 accepted, Ref. 34 in the new manuscript version], resulting in quite high values of 5000-7500 cm²V⁻¹S⁻¹. Therefore, although we cannot rule-out a possible increase of losses for plasmons in 3D nanoporous graphene, we mainly ascribe the broad linewidth Γ_{pl} in NPG to the statistical distribution of Fermi energy and pore-size in the 3D structure.

This argument can be substantiated by calculating the upper and lower plasmon frequency for samples in Table 1 SI due to their Fermi energy and pore-size distribution.

Look at Equation 2 of the main manuscript, the plasmon frequency depends, besides constants, on the ratio $\sqrt{E_F/p}$. This means that the upper (lower) plasmon frequency can be obtained when E_F takes its higher (lower) value and p its lower (upper) value, respectively. By putting these numbers in Equation 2, one obtains for instance for the NPN3 sample $\nu_{pl}^{max}=1130 \text{ cm}^{-1}$ and $\nu_{pl}^{min}=390 \text{ cm}^{-1}$. These frequencies are represented by black dashed vertical lines in Fig. 1 of this Reply. As evident from the Figure the lower and upper frequency limits take into account the broadening of the plasmon band. The same calculation performed for the others samples in Table 1 (not shown), strongly support the same interpretation.

We have changed the main text to explain the large linewidth observed as follows:

The plasmon frequency ν_{pl} vs. E_F , as obtained from fitting of data in Fig. 2, is reported in Fig. 3 (red squares). The error bars have been estimated through a statistical analysis on several samples having, nominally, the same properties (see Supplementary Note 2). ν_{pl} clearly increases with E_F passing from nearly 60 cm^{-1} for $E_F \sim 70 \text{ meV}$ to about 800 cm^{-1} for the highest doped N-NPG sample with $E_F \sim 340 \text{ meV}$. The value of Γ_{pl} also increases from nearly 200 to about 2500 cm^{-1} with increasing E_F , and is much larger than the mean values found in 2D graphene [4]. Let us notice, that the mobility of Dirac carriers in the same NPG samples have been recently measured in Ref. 31 providing quite good values ranging from 5000 to $7500 \text{ cm}^2 \text{ V}^{-1} \text{ s}^{-1}$. Therefore, although we cannot completely rule-out a possible increase of losses in NPG plasmons with respect single-layer graphene, we mainly ascribe their broad linewidth Γ_{pl} to the statistical distribution of Fermi energy and pore-size in the 3D structure).

Point 3) Minor: caption Fig 1c mentions black arrow but panel shows red arrow.

Reply 3

This has been fixed in the caption of Fig.1.

In the inset of panel (c) we show schematically a Dirac interband transition (red arrow).

Reply Reviewer 2

Point 1) The authors compare their structure to monolayer graphene. However the geometry they have is more closely resembles amorphous carbon films; i.e. random graphite. The thickness of the films are $\sim 10 \text{ um}$ in which case it is highly unlikely that all the “graphene” monolayers making up the thick film are electronically decoupled and behave optically and electronically as independent

monolayers.

Reply 1

We respectfully disagree with the reviewer's opinion. Our samples are formed by crystalline high-quality monolayer graphene layers well preserving the 2D graphene properties as reported in Ref. 13: *Angew. Chem. Int. Ed.* (2014, 126, 4822) and *Advanced Materials*, 2016 DOI: 10.1002/adma.201601067 (Ref. 34 in the new manuscript version), which have been appropriately cited in the current manuscript. Indeed, the electronic density of state in the monolithic nanoporous graphene sheet is quite different from graphite and exactly demonstrates the presence of mass-less Dirac Fermions as observed in 2D graphene. Moreover, transistor based on nanoporous graphene (Ref. 34 in the main manuscript), also demonstrated a double layer capacitance which is associated to Dirac Fermion like in 2D graphene. These experimental results prove that our NPG samples are "2D graphene with 3D porous structures", i.e. totally different from *amorphous carbon films*. Additionally, the 10 μm thickness nanoporous graphene built by a monolithic single "graphene" layer shows 99% porosity. It means the spatial distance between graphene layers is over 100-300 nm at minimum, and therefore it can be considered that the nanoporous graphene layers are randomly distributed in 3D space without seams and with/without weak electric interfaces. This clearly suggests that graphene monolayers forming thick samples are electronically decoupled and behave as independent sheets.

Point 2) The structures are completely random leaving many geometric parameters hard to determine. Although the authors perform electron microscopy to determine the pore sizes, the areas characterized in the IR spectrometer can be very different. These two measurements are very difficult to co-localized given the complicated setups required for both techniques.

Reply 2

The main geometry parameter defining the NPG 3D structure is the average porosity p . A measure of the porosity was obtained with the Barrett-Joyner-Halenda (BJH) method [13-16] and confirmed by a direct inspection of SEM images at different positions on the sample surface and separated until hundreds of microns. All infrared measurements have been performed through a spot of 2 mm (which assures to average over any possible spatial disomogeneity), covering the previous inspected regions. Spatial variations of the material response are further investigated with infrared spectromicroscopy. To support this point we have added the following sentence to the Supporting Information:

We also performed infrared transmission measurements through an infrared microscope focusing the radiation over a 35 microns spot. The comparison among macroscopic IR data (2 mm spot) and microscopic data (35 microns spot) for different microscopic points inside the macroscopic spot (see

Fig. 3a SI) and separated of several hundreds of microns is reported in Fig.3b SI. This Figure shows a very good reproducibility and substantial independence of IR spectra from sample surface position (numbered by 1,2, 3 in Fig.3a SI), at least down to micrometric spatial scale.

Fig. 3 SI. a) Image of the NPN5 sample (inset shows the sample mounted free standing on its copper frame). b) Extinction spectra of the same sample taken with a 2 mm spot (black line) and collected through a Bruker Infrared microscope Hyperion 3000 coupled with a Vertex 70V and equipped with a 36 objective at different spatial positions (red, blue and green lines). These positions are indicated in the optical image a). The extinction spectrum for the 2 mm spot is shifted by a multiplicative factor 1.02 for a sake of comparison.

Let us finally observe that Raman microscopy data with a 20 microns spot acquired onto different sample surface points (separated until several mm), confirm the same spatial uniformity of NPG samples (see Ref. 13 and 34 main manuscript).

Point 3) *The most troubling aspect is that all the conclusions are based on fitting the experimental results based on some mathematical model that depend on many parameters such as the pore size, the Fermi level, the film thickness etc. All of these parameters may vary significantly from one area of the sample to another due to the random nature of this specific material system. It is not immediately obvious to this reviewer as to why the fits are unique given that there maybe another set of parameters to produce the same results. In fact it is also not clear why the particular mathematical form is used for the fits. In fact in line 193 they argue that a Gaussian can be used instead of the Lorentzian they adapted.*

Reply 3

We thank the Reviewer for evidentiating this point which, probably, is not clearly explained in the main manuscript especially referring to line 193 (old version).

In order to discuss point 3 we resume briefly the procedure of fitting and data analysis.

a) For any value of average doping and pore size, we measured several samples both with macroscopic and microscopic (35 micron, see Fig. 3 SI) spot size showing a substantial reproducibility and spacial uniformity of optical data. This means that samples with the same doping and pore size are essentially homogeneous down to the tens of microns scale (see Reply 2).

b) From extinction data we extracted the optical conductivity by a Kramers-Kronig consistent fit as usually made in single and few layers graphene.

c) In order to extract the plasmon frequency and Fermi energy, Eq. 5 of SI has been fitted to the real part of the optical conductivity. This Equation has been obtained by convolving the second term of Eq.1 (main text) with a statistical distribution of Fermi energy (see SI). The convolution doesn't refer to the plasmonic absorption which is assumed (as usual) Lorentzian, but only to the interband transition (this point has been clarified in the main text). By using a flat distribution for the Fermi energy, characterized by the extremal values E_{F1} and E_{F2} , the convolution integral can be analytically calculated, producing the third term in Eq. 5 SI. Let us observe, however, that fitting results are practically independent of the Fermi energy statistical distribution. Indeed, one obtains the same results by changing the flat distribution with a Gaussian one.

The need to introduce a statistical distribution of Fermi energy is due to the spatial distribution of dopants in 3D NPG. Thus, the main physical parameters entering in Eq. 5 of SI are the Fermi energy mean value $E_F=(E_{F1}+E_{F2})/2$ and the characteristic plasmon frequency and linewidth, ν_{pl} and Γ_{pl} .

d) The fit results (E_F , ν_{pl} and Γ_{pl} , which are represented in Table S1) do not depend of the specific choice of the statistical distribution function used for the interband contribution. In fact in line 193 of the manuscript (old version), we wrote that changing the flat distribution (and not a Lorentzian as erroneously reported by the Reviewer), with a Gaussian distribution, fitting results are practically the same.

This means, in our opinion, that the whole fitting procedure determines a well defined set of physical parameters characterizing the low-energy electrodynamics of 3D NPG. In particular, the plasmon frequency, whose behavior versus E_F and p , is investigated in Fig. 3 and 6 of the main manuscript are absolutely “robust”, *i.e.* independent of the Fermi energy distribution, and the error bars reported in these figures take into account both the statistical fit uncertainty and sample-to-sample variation of their optical properties (see SI). We observe, finally, that the plasmon frequency behavior versus E_F and $1/p$ is fully consistent with localized Dirac plasmon.

We change now the main manuscript as follow:

Let us observe that the modelling of the interband contribution is robust against the choice of the Fermi energy distribution function.

Point 4) *The authors claim that the resonances originate from the pores in a conducting “graphene” layer implying that they emerge as localized surface plasmon resonances. It is not clear in the text what they mean by the term plasmon. Is it volume plasmons, surface plasmons, or localized surface plasmons? If it is the first, then it is not surprising since any conducting material including metals and doped semiconductors behave as a good reflector below the plasma frequency. If it is surface plasmons—a mode supported at the graphite and air interface, usually they cannot be excited from free space due to the momentum mismatch. If it is localized surface plasmons, the pores are 3D in nature and vary significantly from area to area and layer to layer on the samples based on the SEM images.*

Reply 4

As pointed out we could have been more explicit in our terminology. Therefore we have changed the main text specifying that we are investigating *localized surface plasmons* in Nanoporous Graphene:

This peak, which has been observed also in single-layer disordered graphene [21], will be attributed (see below) to localized surface plasmons in the underlying graphene layers, where the localization and extra wavevector are provided by the nano-scale porosity.

As already discussed in Reply 2 and 3 local variability of the pore size from area to area does not invalidate our study since it is averaged out in the infrared response of the material both at the micro- (35 microns) and macro-scale (2 mm) of the infrared measurement, while the analysis of the spectral features in this investigation gives insight into the underlying inhomogeneity.

While the large infrared spot size in transmission mode averages over thousands of pores, ruling out local variation contributions, the assignment of THz and IR peaks to localized surface plasmon is supported by the observation of the $1/p$ dependence and Fermi energy dependence, which are the fingerprints of localized Dirac plasmons.

Point 5) *Even if we were to assume that the monolayers of graphene in the sample were electrically decoupled, optically they are only less than nm apart and strongly coupled. This is expected to affect the spectral position of the localized surface plasmon resonances significantly. In this case it is not clear why a model based on monolayer is adapted.*

Reply 5

We used a single-layer based model to compare the experimental plasmon frequency behavior versus E_F and $1/p$ with a theoretical dependence (Eq. 2 in the main manuscript), because the average inter-layer distance is over 100-300 nm. This render effectively electrically decoupled the layers in the 3D structure. Equation 2 has been achieved from a theoretical single-hole polarizability in Ref. 35 (main manuscript), and this polarizability has been shown to well reproduce the optical behavior of single-layer graphene hole-decorated surface for different hole diameters, distances and lattice symmetries, including possible hole-hole interaction.

Actually, the plasmon-plasmon interaction has been also experimentally studied in single and multiple-layer hole-decorated graphene [see for instance Hugen Yan et al, New Journal of Physics 14 (2012) 125001]. In a single-layer, already for a interhole distance $d \geq 2p$ (where p is the hole diameter), plasmon interaction (due to its dipolar nature), scarcely affects the plasmon frequency which can be described in terms of Eq. 2 (of the main text).

In multiple layers, a critical parameter is the interlayer distance t . For $t < p$, Hugen Yan et al. found a strong hardening of the plasmon frequency for an increasing number of layers. This is due to a strong hole-hole interaction in adjacent layers determining an increase of the plasmon restoring force and then of the plasmon frequency. Actually, in 3D NPG graphene, due to its random nature, one reasonable expects a reduction of this effect. Indeed, the interlayer distance should be larger than the pore size (reducing the dipolar interaction) and the statistical distribution of pores in adjacent layers also reduces their effective interaction.

In conclusion, we expect that a single pore description should describe (as experimentally demonstrated in Fig. 4 and 5 of the main manuscript) the behavior of localized Dirac plasmons in 3D NPG graphene.

Reply Reviewer 3

Point 1) the abstract should be further strengthened, particularly highlighting the impact of this study for applications in the midinfrared. Unlike graphene plasmonics, here the 3D nature implies that the plasmonic effect is polarization insensitive and can be used more broadly in certain type of applications.

Reply 1

We thank Reviewer 3 for this helpful insight and we have thus expanded the final part of the abstract as follows:

We reveal the presence of intrinsic 2D Dirac plasmons in 3D nanoporous graphene disclosing strong plasmonic absorptions tunable from terahertz to mid-infrared via controllable doping level and porosity. The observed broad plasmon linewidths allow to cover most of the molecular mid-infrared fingerprint region with a single plasmon excitation. The enhanced surface area of nanoporous structures combined with their broad band plasmon absorption could pave the way for novel and

competitive nanoporous-graphene based plasmonic-sensors.

And modified the conclusions:

Furthermore, for N-doped samples, the microscopic inhomogeneity of doping and pore sizes yields a macroscopic plasmonic response that cover a wide spectral range ($>1000\text{ cm}^{-1}$). By taking into account the enhanced surface area of nanoporous structures, the tunability of graphene plasmon and the broad spectral response, the use of 3D nanoporous graphene could pave the way for novel and competitive graphene based plasmonic-sensors.

Point 2) would the plasmon to plasmon hybridization be important in the analysis of the plasmon resonance. I would expect this to be sensitive to the filling factor. In other words, what is the pores volume to the total volume.

Reply 2

We used a single-layer based model to compare the experimental plasmon frequency behavior versus E_F and $1/p$ with a theoretical dependence (Eq. 2 in the main manuscript). This Equation has been achieved from a theoretical single-hole polarizability in Ref. 35 (main manuscript), and this polarizability has been shown to well reproduce the optical behavior of single-layer graphene hole-decorated surface for different hole diameter, distance and lattice symmetry, including possible hole-hole interaction.

Actually, the plasmon-plasmon interaction has been also experimentally studied in single and multiple-layer hole-decorated graphene [see for instance Hugen Yan et al, New Journal of Physics 14 (2012) 125001]. In a single-layer, already for a interhole distance $d \geq 2p$ (where p is the hole diameter), plasmon interaction (due to its dipolar nature), scarcely affects the plasmon frequency which can be described in terms of Eq. 2 (of the main text).

In multiple layers, a critical parameter is the interlayer distance t . For, $t < p$, Hugen Yan et al. found a strong hardening of the plasmon frequency for an increasing number of layers. This is due to a strong hole-hole interaction in adjacent layers determining an increasing of the plasmon restoring force and then of the plasmon frequency. Actually, in 3D NPG graphene, due to its random nature, one reasonable expects a reduction of this effect. Indeed, the interlayer (effective) distance should be larger than the pore size (reducing the dipolar interaction) and the statistical distribution of pores in adjacent layers also reduces their effective interaction.

In conclusion, we expect that a single pore description should describe (as experimentally demonstrated in Fig. 4 and 5 of the main manuscript) the behavior of localized Dirac plasmons in 3D NPG graphene.

Point 3) the observed plasmon line width appears to be quite large. The authors should discuss the source of “damping” or probably this is due to pore size variability.

Reply 3

The observed linewidths are larger than what is found in single-layer based graphene nanostructures, and this is due to both pore size variability and Fermi energy inhomogeneity within the area probed by the IR spot size. To highlight this difference with 2D graphene, we have reported the value of the fitting parameters in Table S1 (see below).

In the first column of this table, NP and NPN are relative to undoped and N-doped samples respectively. The II, III, IV and Vth columns report the pore size p , the plasmon frequency ν_{pl} , their widths Γ_{pl} , the average Fermi energy E_F and the corresponding statistical width.

Let us note that data reported in Table S1 correspond to an average on several samples having, nominally, the same physical properties.

Batch ID	p (nm)	ν_{pl} (cm ⁻¹)	Γ_{pl} (cm ⁻¹)	E_F (cm ⁻¹)
NP1	200 ± 50	280 ± 30	440 ± 45	570 ± 150
NP2	900 ± 100	60 ± 5	190 ± 20	580 ± 250
NP3	650 ± 90	150 ± 15	480 ± 50	560 ± 300
NPN1	200 ± 50	450 ± 50	1550 ± 150	1770 ± 600
NPN2	200 ± 50	670 ± 70	2550 ± 250	2100 ± 600
NPN3	200 ± 50	800 ± 80	2450 ± 250	2800 ± 1000

Table 1. Physical parameters for the measured nanoporous samples.

As evident from the Table, NPG plasmons show a large spectral broadening Γ_{pl} . This is actually larger than linewidth related to single layer graphene plasmon located at similar central frequency. For instance, the low-doped NP2 sample shows a THz plasmon centered around 150 cm⁻¹ with a spectral width Γ_{pl} of about 480 cm⁻¹. THz plasmons in single-layer graphene decorated with disk or ribbon array, with a similar central frequency, presents a linewidth around 100 cm⁻¹ [see Ref. 4 of the main manuscript and Yan et al., Nature Nanotech. 7, 330 (2012)]. A similar overdamped behavior can be observed in Table S1 also for mid-IR plasmons.

In our opinion, this behavior is not related to an intrinsic low-mobility of Dirac carriers in 3D NPG graphene. Indeed, the carrier mobility has been recently measured in similar NPG samples [Y. Tanabe et al., “Electric Properties of Dirac Fermions Captured into 3D Nanoporous Graphene Networks” Advanced Materials, accepted 2016, Ref. 34 in the new manuscript version], resulting in good values of 5000-7500 cm²V⁻¹s⁻¹. Therefore, although we cannot rule-out a possible increase of losses for

plasmons in 3D nanoporous graphene, we mainly ascribe the broad linewidth Γ_{pl} in NPG to the statistical distribution of Fermi energy and pore-size in the 3D structure.

This argument can be substantiated by calculating the upper and lower plasmon frequency for samples in Table 1 SI due to their Fermi energy and pore-size distribution.

Look at Equation 2 of the main manuscript, the plasmon frequency depends, besides constants, on the ratio $\sqrt{E_F/p}$. This means that the upper (lower) plasmon frequency can be obtained when E_F takes its higher (lower) value and p its lower (upper) value, respectively. By putting these numbers in Equation 2, one obtains for instance for the NPN3 sample $\nu_{pl}^{\max}=1130 \text{ cm}^{-1}$ and $\nu_{pl}^{\min}=390 \text{ cm}^{-1}$. These frequencies are represented by black dashed vertical lines in Fig. 1 of this Reply. As evident from the Figure the lower and upper frequency limits fairly take into account the broadening of the plasmon band. The same calculation performed for the others samples in Table S1 (not shown), strongly support the same interpretation.

Therefore, and we have changed the main text as follows:

The plasmon frequency ν_{pl} vs. E_F as obtained from fitting of data in Fig. 2, is reported in Fig. 3 (red squares). The error bars have been estimated through a statistical analysis on several samples having, nominally, the same properties (see Supplementary Note 2). ν_{pl} clearly increases with E_F passing from nearly 60 cm^{-1} for $E_F \sim 70 \text{ meV}$ to about 800 cm^{-1} for the highest doped N-NPG sample with $E_F \sim 340 \text{ meV}$. The value of Γ_{pl} also increases from 200 to 2500 cm^{-1} with increasing E_F , and is much larger than the mean values found in 2D graphene [4]. Let us notice, that the mobility of Dirac carriers in the same NPG samples have been recently measured in Ref. 34 providing quite good values ranging from 5000 to $7500 \text{ cm}^2 \text{ V}^{-1} \text{ s}^{-1}$. Therefore, although we cannot completely rule-out a possible increase of losses in NPG plasmons with respect single-layer graphene, we mainly ascribe their broad linewidth Γ_{pl} to the statistical distribution of Fermi energy and pore-size in the 3D structure.

Point 4) *the data set in figure 3 and 5 is quite scarce, where most of the data points were taken from other published work. I feel more experiment data should be included in figure 3 and 5. With more data, they can also analyzed the plasmon line width and analyze its dependence with pore size. Line width should increase with decreasing pore size, per what has been reported in nanoribbons array.*

Reply 4

The Fermi energy range measured in this work and represented in Fig. 3 spans from nearly 70 to 340 meV which is comparable to most of data present in literature. For what concerns pore size data in

Fig. 5, we sampled the whole range from $p \sim 200$ nm to 1 microns actually accessible to sample growing technique. Moreover, the plasmon frequency data set shown both in Fig. 3 and 5 come from an average on several NPG samples having, nominally, the same physical properties (same Fermi energy and pore size). Actually, up to four samples *per* plasmon frequency data are reported in these Figures. On this basis, we think that the NPG data set measured in this work represents the (actually) best description of plasmon behavior versus l/p and E_F in nanoporous graphene samples. Possible extension of data at lower pore size (below 200 nm) and higher Fermi energy (above 350 meV), if permitted by an improvement of sample growing, will be reported in future papers.

Finally, as reported in Table S1, the plasmon linewidth depends both on the Fermi energy and pore size. In particular, for a fixed $E_F \sim 70$ meV, the linewidth increases with l/p , as observed in single layer graphene (see Reply 3).

REVIEWERS' COMMENTS:

Reviewer #1 (Remarks to the Author):

The authors have answered all the questions very clearly and modified the manuscript such that the quality clearly improved. The work is of high standard and for sure deserves rapid publication in Nature Communications.

Reviewer #2 (Remarks to the Author):

The authors have addressed some of the comments in detail and in most cases in a satisfactory fashion. Although it is appreciated that they have gone in length to address the comments for the original submission, this reviewer still thinks that the manuscript offers only a incremental advance compared to the state-of-the-art in graphene plasmonics. The field of graphene plasmonics based on monolayer material is well-established and localized surface plasmons in such configurations have already been investigated in great detail. On the other hand, this revised manuscript extends some of these ideas to the nanoporous graphene material system. Based on the presented data, the plasmon resonances in this 3D configuration offers lower quality resonances not providing any promise for applications over monolayer graphene. Authors emphasize stronger optical contrast due to the thicker material involved in their case. However, in most cases sufficient signal to noise ratio can be achieved in monolayer graphene plasmonics. Although I find it commendable that the authors have studied the plasmonic properties of nanoporous graphene in such detail, I do not believe that the findings of this paper will be of interest to the broader scientific community or bring new avenues of thinking to an already crowded field of graphene plasmonics beyond the use of a different configuration. Therefore, this reviewer is not convinced that the revised manuscript will warrant publication in this journal.

Reviewer #3 (Remarks to the Author):

The authors have addressed my concerns and I recommend the publication of their work in Nature Comm. I would request they include the extensive discussion on damping in the SI.

Reviewer #1

The authors have answered all the questions very clearly and modified the manuscript such that the quality clearly improved. The work is of high standard and for sure deserves rapid publication in Nature Communications.

Reply.

We thank Reviewer #1 for suggesting to publish our paper on Nature Communications.

Reviewer #2

The authors have addressed some of the comments in detail and in most cases in a satisfactory fashion. Although it is appreciated that they have gone in length to address the comments for the original submission, this reviewer still thinks that the manuscript offers only a incremental advance compared to the state-of-the-art in graphene plasmonics. The field of graphene plasmonics based on monolayer material is well-established and localized surface plasmons in such configurations have already been investigated in great detail. On the other hand, this revised manuscript extends some of these ideas to the nanoporous graphene material system. Based on the presented data, the plasmon resonances in this 3D configuration offers lower quality resonances not providing any promise for applications over monolayer graphene. Authors emphasize stronger optical contrast due to the thicker material involved in their case. However, in most cases sufficient signal to noise ratio can be achieved in monolayer graphene plasmonics. Although I find it commendable that the authors have studied the plasmonic properties of nanoporous graphene in such detail, I do not believe that the findings of this paper will be of interest to the broader scientific community or bring new avenues of thinking to an already crowded field of graphene plasmonics beyond the use of a different configuration. Therefore, this reviewer is not convinced that the revised manuscript will warrant publication in this journal.

Reply.

We do not agree with Reviewer #2.

In this paper we have shown that low-energy collective excitations in naturally nanostructured 3D graphene are Dirac plasmons. This is a new and, in our opinion, an important result, which is absolutely not expected in a so strongly disordered material. This result has been demonstrated, by

measuring several nanoporous graphene samples, versus doping and pore size and showing that the plasmon frequency exactly follows the 2D Dirac dispersion. We believe that this finding will be of interest to the broader scientific community, in the emergent field of 3D graphene.

Reviewer #3

The authors have addressed my concerns and I recommend the publication of their work in Nature Comm. I would request they include the extensive discussion on damping in the SI.

Reply.

We thank Reviewer #3 for suggesting to publish our paper on Nature Communications. We have included the discussion on damping in the Supplementary Information.